# Seleno-Analogs of Scaffolds Resembling Natural Products a Novel Warhead toward Dual Compounds

**DOI:** 10.3390/antiox12010139

**Published:** 2023-01-06

**Authors:** Nora Astrain-Redin, Irene Talavera, Esther Moreno, María J. Ramírez, Nuria Martínez-Sáez, Ignacio Encío, Arun K. Sharma, Carmen Sanmartín, Daniel Plano

**Affiliations:** 1Departamento de Tecnología y Química Farmacéuticas, Facultad de Farmacia y Nutrición, Universidad de Navarra, Irunlarrea 1, E-31008 Pamplona, Spain; 2Departamento de Farmacología y Toxicología, Facultad de Farmacia y Nutrición, Universidad de Navarra, Irunlarrea 1, E-31008 Pamplona, Spain; 3Departamento de Ciencias de la Salud, Universidad Pública de Navarra, Avda. Barañain s/n, E-31008 Pamplona, Spain; 4Department of Pharmacology, Penn State Cancer Institute, CH72, Penn State College of Medicine, 500 University Drive, Hershey, PA 17033, USA

**Keywords:** cancer, acetylcholinesterase, selenium, NSAIDs, allyl, propargyl, Alzheimer’s disease, garlic, selenoester, natural

## Abstract

Nowadays, oxidative cell damage is one of the common features of cancer and Alzheimer’s disease (AD), and Se-containing molecules, such as ebselen, which has demonstrated strong antioxidant activity, have demonstrated well-established preventive effects against both diseases. In this study, a total of 39 Se-derivatives were synthesized, purified, and spectroscopically characterized by NMR. Antioxidant ability was tested using the DPPH assay, while antiproliferative activity was screened in breast, lung, prostate, and colorectal cancer cell lines. In addition, as a first approach to evaluate their potential anti-Alzheimer activity, the in vitro acetylcholinesterase inhibition (AChEI) was tested. Regarding antioxidant properties, compound **13a** showed concentration- and time-dependent radical scavenging activity. Additionally, compounds **14a** and **17a** showed high activity in the melanoma and ovarian cancer cell lines, with LD_50_ values below 9.2 µM. Interestingly, in the AChEI test, compound **14a** showed almost identical inhibitory activity to galantamine along with a 3-fold higher in vitro BBB permeation (Pe = 36.92 × 10^−6^ cm/s). Molecular dynamics simulations of the aspirin derivatives (**14a** and **14b**) confirm the importance of the allylic group instead of the propargyl one. Altogether, it is concluded that some of these newly synthesized Se-derivatives, such as **14a**, might become very promising candidates to treat both cancer and AD.

## 1. Introduction

Cancer and Alzheimer’s disease (AD) entail a socio-economic burden for society, accounting for almost 10.0 million deaths, presenting 19.3 million new cases diagnosed in 2020 for cancer [1] and at least 50 million people living with AD or other dementias worldwide [2]. Although treatment options for both illnesses have increased over the past few years, all of them present important drawbacks such as toxicities and resistance. Thus, the development of novel and more effective agents for both diseases is an urgent need.

Recently, miRNAs analysis has confirmed that both diseases present mutual biological hallmarks, including autophagy, the ubiquitin-proteasome system, and cell death. Albeit common features, these pathways might be modulated differently and sometimes in opposite directions [3]. This different modulation, among others, might be the reason why there are no dual drugs in the pipeline. However, few pre-clinical investigations have identified dual agents, including tacrine [4], *N*-aryloxazol-2-amines [5], roscovitine [6], and olaparib [7] analogs, targeting different pathways. Drug discovery toolkits (DDTs) would boost the development of novel compounds with antiproliferative and anti-Alzheimer dual activity. Among the different approaches used to design novel anticancer or anti-Alzheimer compounds, the following ones have demonstrated promising results: (i) computer-aided design, including machine learning techniques; (ii) repurposing; (iii) fragment-based design and molecular hybridization; (iv) natural sources and total synthesis; (v) modified antibodies, and (vi) block copolymer prodrugs [7,8,9,10,11,12].

Overall, considering all the above mentioned, herein we decided to combine several of the DDTs to design “pseudo-natural products” with the potential to become dual agents towards cancer and AD. Thus, the designed compounds gather in the same structure a natural occurring functionality modified with the selenium (Se) atom following a fragment-based design. After an extensive review of the literature, we rule to choose allylic and propargylic fragments present in active molecules of natural products [13,14] (see Figure 1). Molecules containing allylic fragments are present in garlic, nutmeg, parsley, or mustard [15] and active molecules containing propargylic fragments have been identified in cyanobacteria and marine mollusks [16].

Numerous extensive literature review papers have compiled many of the pharmacological effects of garlic extracts and some of its allyl sulfides [17,18,19,20,21,22,23], including preclinical, in vivo, and clinical trials. Two of the most characterized allylsulfides are allicin and *S*-allylcysteine (depicted in Figure 1). The first one possesses anticancer activity through the STAT3 signaling pathway [24,25] along with its anti-amyloidogenic property that prevents the progression of AD [26,27]. Notwithstanding, *S*-allylcysteine has suppressed ovarian cancer proliferation [28]. On the other hand, the propargyl group has been broadly exploited since its privileged structural feature for targeting a wide range of therapeutic target proteins, including MAO or tyrosine kinases. Moreover, propargyl compounds have attracted great interest due to their wide application in organic synthesis as well as the development of active propargyl molecules such as erlotinib and noreynodrel [29,30]. A common feature among these garlic active principles, depicted in Figure 1, is the presence of thioallyl functionality. Accordingly, we elected to replace sulfur by Se on this functionality, given that different selenocompounds have demonstrated anticancer [31,32,33,34,35,36] and anti-Alzheimer [37,38,39] activities.

Se is a trace mineral, normally found in selenoproteins, and it is essential for the proper working of the human organism. Recently, the Se atom has gained attention because of its antioxidant capacity and its great versatility, making it the starting point for numerous studies in search of new active molecules [40]. One of the common features of cancer and AD is oxidative cell damage. In AD, oxidative stress is related to the early stage of the disease, prior to cytopathology, and it enhances the development of the disease [41]. On the other hand, cancer cells produce elevated levels of ROS, which is required for early tumor development and late tumor progression [42]. Additionally, Se-containing molecules, such as ebselen, demonstrated strong antioxidant activity [43]. Ebselen has been demonstrated to reverse peripheral oxidative stress induced by a mouse model of sporadic AD [39]. On the other hand, the relationship between Se and cancer has been thoroughly studied and depends on Se concentration, with high levels being cytotoxic. Likewise, the chemical form in which Se derivatives are administrated and their metabolism play a relevant role in their anticancer activity [44,45,46]. Nevertheless, the study of the relationship between Se and AD is relatively recent. Se has been demonstrated to be crucial for brain functions, and severe Se deficiency causes irreversible brain injury [47]. Se exerts its biological functions mainly through selenoproteins, which are characterized by containing a selenocysteine amino acid residue, and they play a crucial role in antioxidant defense. The emerging evidence of the potential of Se for the prevention and treatment of AD has led to great interest in the study of the most abundant selenoproteins in the brain. One of them is selenoprotein P, which is highly expressed in the brain due to its Se transport function [48]. It was demonstrated that mice lacking the machinery for selenoprotein P synthesis under Se-deficient conditions developed spasticity, abnormal movements, and seizures [49]. In addition, selenoprotein P has been observed to inhibit the Cu-induced Aβ aggregation in vitro [50]. Moreover, selenate and diphenyldiselenide (PhSe)_2_ have been demonstrated to decrease Aβ accumulation by decreasing the activity of β-secretase and γ-secretase [51,52]. Similarly, selenate has been shown to decrease tau hyperphosphorylation and NFT formation by activating serine protein phosphatase 2A. In this study, it was observed that the administration of selenate to rodents produced cognitive improvements and reduced neurodegeneration. This observation could be related to the transformation of selenate to specific selenoproteins including glutathione peroxidase that could attenuate intracellular ROS [53]. Given these results, new selenocompounds were synthesized and shown to inhibit the enzyme acetylcholinesterase and the enzyme BACE-1, both of which are implicated in AD [54,55]. Recently, most of the organic selenoderivatives have turned out to be multi-target compounds, a characteristic that points toward the incorporation of Se as an ideal approach to developing novel antiproliferative and anti-Alzheimer dual agents. Furthermore, it is well-established that the incorporation of Se atom into organic molecules can achieve an exponential increase in the bioactivity of the parent compounds in several diseases [56,57,58]. On the other hand, non-steroidal anti-inflammatory drugs (NSAIDs) are characterized by the inhibition of the enzyme cyclooxygenase (COX), involved in inflammatory processes and prostanoid signaling. In cancer, chronic inflammation plays a key role, and extensive research associates inflammatory cells in the tumor microenvironment (TME) with tumor initiation, cellular proliferation, and local invasion [59,60]. Indeed, in many cases, cancer-promoting inflammation is induced and exists early on or even before tumor formation, as in inflammatory bowel disease (IBD) [61]. The isoenzyme COX-2 is related to some epithelial cancers such as breast cancer [62]. Thus, the antiproliferative effects of NSAIDs have been investigated against several cancer cell lines. For instance, a series of studies reported the efficacy of aspirin in improving breast and colorectal cancer survival [63,64,65,66]. Likewise, the possible link between AD and inflammation has recently been studied and the crucial role of COX-2 isoenzyme for β-amyloid protein propagation and reduction of tau glycosylation in AD has been established [67,68]. Therefore, scientific studies have been carried out to evaluate the effect of NSAIDs in AD and have demonstrated their capacity to inhibit Aβ aggregation, such as indomethacin via a α2-macroglobulin-activating lrp1-dependent mechanism [69,70,71]. Thus, the development of novel compounds that include NSAIDs in their structure seems to be a promising approach in the search for dual drugs against cancer and AD.

Based on the above facts, the current study presents the synthesis and in vitro evaluation of 38 Se-containing compounds as dual anticancer and anti-Alzheimer agents (see Figure 2). These Se derivatives were designed based on a fragment-based approach, gathering, in the same molecule, two active fragments and the Se atom, in the form of selenoester. In the acyl moiety, we envisioned the introduction of different small carbo- and hetero-cycles, as well as NSAIDs. In the opposite location of the molecule, allyl (**series a**) or propargyl (**series b**) fragments were included with the aim of mimicking the active ingredients of natural products. To date, the library of Se-containing compounds is extensive due to its high chemical versatility, and it is greatly increasing yearly [72]. There are molecules with acylselenourea and selenourea groups in their structures that have been found to be excellent radical scavenger agents. Within these groups of Se derivatives, molecules with dual in vitro antioxidant and antiproliferative activities have been also reported [36,73]. Moreover, selenoester derivatives have been observed as potent cytotoxic agents, albeit not all of them exhibit antioxidant activity [74,75]. The most studied Se-containing compound is ebselen, with a benzoisoselenazolone ring, which has exhibited anti-inflammatory, antioxidant, and anticancer activities [43,76,77,78]. However, no molecules have been reported so far that combine in their structure NSAIDs and an active fragment present in garlic-derived natural products through a selenoester group. All the synthesized Se derivatives were evaluated as dual agents towards breast, colorectal, lung, and prostate cancer and AD using the MTT assay and Ellman’s method, respectively. Moreover, DPPH assay was performed to evaluate the antioxidant capacity of the synthesized compounds. The two most active compounds in the screening against cancer cell lines were submitted to the Developmental Therapeutics Program (DTP) of the National Cancer Institute (NCI). Furthermore, molecular dynamics simulations were carried out to elucidate the mode of interaction between compound **14a** and the active site of AChE.

## 2. Materials and Methods

### 2.1. Chemistry

#### 2.1.1. General Remarks

NMR spectra were recorded on a Bruker Avance Neo 400 MHz using Chloroform-d as a solvent. Chemical shifts were reported in δ values (ppm) and J values were reported in hertz (Hz). All the reaction procedures were monitored by Thin Layer Chromatography (TLC) using Alugram® SIL G/UV254 sheets (Layer: 0.2 mm) (Darmstadt, Germany). TLC were visualized by exposure to ultraviolet light. Final products were obtained by column chromatography using Silica gel 60 (0.040–0.063 mm) (Merk KGaA, Darmstadt, Germany). Chemicals were purchased from E. Merck (Darmstadt, Germany), Panreac Química S.A. (Montcada i Reixac, Barcelona, Spain), Sigma-Aldrich Química, S.A. (Alcobendas, Madrid, Spain), Acros Organics (Janssen Pharmaceuticalaan 3a, 2440 Geel, Belgium) and Lancaster (Bischheim-Strasbourg, France). Melting points (mp) were determined with a Mettler FP82+FP80 apparatus (Greifensee, Switzerland). All the compounds are >95% pure by quantitative NMR (^1^H q-NMR) using dimethyl sulfone as reference. The results are expressed as the percentage of purity and were calculated tracking the signal of the first alkene hydrogen, which appears around 5.9 ppm, for **series a**, and the signal of CH_2_ which appears around 3.7 ppm, for **series b**.

#### 2.1.2. General Procedure for the Synthesis of the Compounds (**1a**–**19a** and **1b**–**19b**)

NaBH_4_ (0.9 g, 11.39 mmol) was added to a mixture of elemental Se (0.9 g, 11.39 mmol) in water (30 mL). The mixture was stirred at room temperature for 20 min. The corresponding acyl chloride (11.39 mmol) was added in situ and stirred for 60 min. Allyl bromide (0.9 mL, 11.39 mmol) for **series a**, and propargyl bromide (0.8 mL, 11.39 mmol) for **series b**, and tetrahydrofuran (10 mL) were added in situ and stirred at room temperature for 90 min. The product was isolated by extraction with methylene chloride (3 × 50 mL) and the organic phases were dried using sodium sulfate anhydrous. After that, the organic phases were filtered, and the methylene chloride was removed by rotatory evaporation under vacuum. The final product was purified by column chromatography. A gradient of hexane/ethyl acetate, ranging between 0% and 50% of ethyl acetate, was used as eluent. The mobile phase of the TLCs was hexane/ethyl acetate with a ratio of 9:1. The Rf range was from 0.18 to 0.75.

The acyl chlorides of compounds **1**, **4**, **10**, **7**–**8,** and **15**–**19** were formed from the reaction of the corresponding carboxylic acids with oxalyl chloride in methylene chloride using drops of *N,N*-DMF as a catalyst. The mixture was stirred at room temperature for twelve hours and the reaction media was removed by rotatory evaporation under the vacuum. Table 1 and Table 2 provide the chemical name, starting reagent, yield, appearance, ^1^H, ^13^C, and ^77^Se NMR spectra, and purity data for the compounds of **series a** and **series b**, respectively.

### 2.2. Biology

#### 2.2.1. DPPH Free Radical-Scavenging Assay

The antioxidant activity was determined by the colorimetric assay of DPPH, described by Svinyarov [79], in which the capacity of the compounds for scavenging radicals in vitro is measured. It is based on the reduction of a stable free radical, DPPH, with the presence of antioxidants, by the donation of a hydrogen atom. Then, this reduction of DPPH causes the decrease in its absorbance at 517 nm, and the corresponding DPPH radical-scavenging activity can be determined. The measurements were recorded on a BioTeck PowerWave XS spectrophotometer (BioTeck Instruments, Winooski, VT, USA) and the data were collected using KCJunior v.1.41. software (BioTeck Instruments, Winooski, VT, USA).

Each compound was dissolved in absolute methanol at the concentration of 1 mg/mL, and then, serial dilutions were prepared. Ascorbic acid and trolox were used as positive controls because of their well-known and potent antioxidant capacity. A methanolic solution (0.04 mg/mL) of DPPH (Aldrich) was prepared daily and was protected from light. The blank of colorless sample was absolute methanol. Moreover, 100 μL of each sample were dissolved in 100 μL of DPPH solution, and the control was prepared dissolving 100 μL of absolute methanol in 100 μL of the DPPH solution. The decolorization of the purple radical to the yellow reduced form was followed at 517 nm and the absorbances were read after different times intervals. All the measurements were carried out in triplicate. Results are expressed as the percentage of the radical scavenger, calculated using the following formula:$$\%\;\mathrm{DPPH}\;\mathrm{radical}\;\mathrm{scavenging}=\frac{\left(A_{control\;}-A_{blank}\right)-\left(A_{sample\;}-A_{blank}\right)}{\left(A_{control}-A_{blank}\right)}$$

#### 2.2.2. Cell Culture Conditions

The cell lines were obtained from the American Type Culture Collection (ATCC). Five tumor cell lines (MDA-MB-231, MCF-7, HCT116, HTB-54, and DU-145) and two non-tumorigenic cells (184B5 and BEAS-2B) were grown in RPMI 1640 medium (Gibco), and the HT-29 tumor cell line was grown in McCoy’s medium. Both mediums were supplemented with 10% fetal bovine serum (FBS; Gibco) and 1% antibiotics (10.00 units/mL penicillin and 10.00 μg/mL streptomycin; Gibco). Cells were preserved in tissue culture flasks at 37 ^o^C and 5% CO_2_. The culture medium was replaced every three days.

#### 2.2.3. Cell Viability Assay

The effect of each compound on cell viability was tested using the MTT assay [80]. Each compound was dissolved in dimethyl sulfoxide (DMSO) at a concentration of 0.01 M. At first, the inhibition of cell viability was determined at two different concentrations (10 and 50 µM) in MDA-MB-231, HTB-54, DU-145 and HT-29 cells. Selected compounds were then tested at seven different concentrations (0.5–100 μM) in MCF-7, HCT116, HT-29, HTB-54, 184B5 and BEAS-2B cells. Cells were seeded at 10^4^ per well onto flat-bottomed 96-well culture plates. They were treated with either DMSO or increasing concentration of the corresponding compound for 48 h. Then, they were incubated with 20 μL of 3-(4,5-dimethylthiazol-2-yl)-2,5-diphenyltetrazolium bromide (MTT) (2 mg /mL stock; Aldrich) for 2,5 h and analyzed for their ability to generate a purple formazan dye. These formazan crystals were dissolved in 50 μL of DMSO. The absorbance was measured at a wavelength of 550 nm and the ratio of viable cells was calculated.

Results are expressed as cell viability % at 10 μM. Selectivity index were calculated as the ratio of the IC_50_ values determined for the non-malignant and the tumor cells in breast (IC_50_ (184B5)/IC_50_ (MCF-7)) and lung (IC_50_ (BEAS-2B)/IC_50_ (HTB-54)) cell lines. Data were obtained from at least three independent experiments performed in triplicates.

#### 2.2.4. NCI-60 Analysis

Compounds **14a** and **17a** were submitted to the Developmental Therapeutics Program (DTP) of National Cancer Institute (NCI). Cytotoxicity activity was evaluated by performing One-Dose screening (10^−5^ M) against a panel of 60 human tumor cell lines. As both compounds demonstrated effective cytotoxic activity, they were selected for the Five-Dose (0.01 μM–100 μM) assay against the same cell panel comprising different leukemia, non-small cell lung, colon, central nervous system (CNS), melanoma, ovarian, renal, prostate and breast tumor cell lines. Cells grew in the RPMI 1640 medium containing 5% fetal bovine serum and 2 mM *L*-glutamine and were inoculated into 96-well microtiter plates in 100 μL at plating densities ranging from 5.000 to 40.000 cells/well. The microtiter plates inoculated with cells were incubated at 37 °C, in 5% humidified CO_2_ for 24 h prior to addition of experimental drugs. Then, experimental drugs were added, and the microtiter plates were incubated for 48 h. The protocol is available on https://dtp.cancer.gov/discovery_development/nci-60/methodology.htm accessed on 31 December 2022.

#### 2.2.5. AChE Inhibition

The AChE inhibitory capacity of the synthesized compounds, **1a**–**19a** and **1b**–**19b**, was assessed spectrophotometrically by Ellman’s method [81] with minor modifications [82]. Frontal cortex tissue obtained from male Wistar rats was homogenized in 39 volumes of 75 mM sodium phosphate buffer (pH 7.4). A mixture of 260 μL containing the compounds assessed, acetylthiocholine iodide and 5 μL tissue homogenate was incubated for 8 min. The reaction was then terminated by adding 50 μL 3% (w/v) sodium dodecyl sulphate followed by 50 μL 0.2% (*w*/*v*) 5,5′-dithio-bis(2-nitrobenzoic) acid to produce the yellow anion of 5-thio-2-nitro-benzoic-acid. The extent of the color production was measured spectrophotometrically at 405 nm using Multiskan Ex (Thermo Electron Corporation). Compounds were assessed at a screening concentration of 10^−6^ M. IC_50_ values were calculated as the concentration of compound that produces 50% enzyme activity inhibition with the OriginPro 9.0.0 software. Results are expressed as the mean ± SD and each experiment was repeated three times.

#### 2.2.6. In Vitro Brain–Blood Barrier Permeation Study

The parallel artificial membrane permeability assay (PAMPA) described by Di et al. was used to evaluate the brain penetration of the compounds **10a**, **13a**, **14a**, and galantamine (used as reference drug) [83]. Franz diffusion cells (Microette 8910130, Hanson Research) were employed to perform this assay. The compounds were dissolved in DMSO at 5 mg/mL and diluted to 500 µg/mL with phosphate-buffered saline (PBS)/EtOH (7:3) to make a stock solution. A total of 10 µL of porcine brain lipid (PBL) diluted in dodecane solution (20 mg/mL) was spread onto a PVDF membrane that was placed between the donor and acceptor compartments, forming a sandwich structure. Then, 4.5 mL of PBS/EtOH (7:3) was added to the acceptor compartment and 700 µL of the stock solution was added to the donor cell. After maintaining this structure for 20 h at 25 °C, the donor cell was carefully removed, and the concentrations of the tested compounds in the acceptor and donor cells were measured as UV-visible (λ = 272 nm for compound **10a**, and λ = 290 nm for compound **14a** and **galantamine**) (8453 UV-Visible Agilent Technologies). The concentration of the compound was calculated from the standard curve and expressed as permeability (*Pe*) by the following formula [84]:Permeability (cm/s): *Pe* = {−ln [1 − *C*_A_(*t*)/*C*_eq_]}/[A ∗ (1/*V*_D_ + 1/*V*_A_) ∗ *t*]

A = filter area (0.636 cm^3^), *V*_D_ = donor cell volume (0.7 mL), *V*_A_ = acceptor cell volume (4.5 mL), *t* = incubation time, *C*_A_ (*t*) = compound concentration in acceptor cell at time *t*, *C*_D_ (*t*) = compound concentration in donor cell at time *t*, and *C*_eq_ = [*C*_D_(*t*) ∗ *V*_D_ + *V*_A_(*t*) ∗ *V*_A_]/(*V*_D_ + *V*_A_).

Every sample was analyzed at least in duplicate and the data were reported as mean ± SD.

#### 2.2.7. Statistical Analysis

Data were expressed as the mean ± SD (standard deviation) and experiments were performed three times in triplicates unless otherwise specified. Non-linear curve regression analysis calculated by OriginPro 9.0.0 software (OriginLab Corporation; Northampton, MA, USA) was used to assess the IC_50_. Data were analyzed using GraphPad Prism version 8.0.2 (GraphPad Software; San Diego, CA, USA).

### 2.3. Molecular Dynamics Simulations

The following crystal structures deposited in the Protein Data Bank (PDB) under the ID 4BDT [85] and 4EY6 [86] were used to prepare the starting structure to run the Molecular Dynamics Simulations (MD). The ligand (galantamine) of the 4BDT structure was replaced by the different selenium derivatives before running the MD simulations. Both the system preparation and the simulations were performed in the AMBER 18 suite software. The protocol for the system preparation and the MD simulations is detailed as follows. Firstly, the system is neutralized by adding sodium ions and later immersed in a cubic box of 10 Å length, in each direction from the end of the protein, using TIP3P water parameters. The force fields used to obtain topography and coordinates files were ff14SB [87] and GAFF [88]. The first step of the simulation protocol followed to run the MD simulations is a minimization of the solvent molecules position only, keeping the solute atom positions restrained, and the second stage minimizes all the atoms in the simulation cell. Heating the system is the third step, which gradually raises the temperature 0 to 300 K under a constant volume (ntp = 0) and periodic boundary conditions. In addition, Harmonic restraints of 10 kcal·mol^−1^ were applied to the solute, and the Berendsen temperature coupling scheme [89] was used to control and equalize the temperature. The time step was kept at 2 fs during the heating phase. Long-range electrostatic effects were modelled using the particle-mesh-Ewald method [90]. The Lennard-Jones interactions cut-off was set at 8 Å. An equilibration step for 100 ps with a 2 fs time step at a constant pressure and temperature of 300 K was performed prior to the production stage. The trajectory production stage kept the equilibration step conditions and was prolonged for 500 ns longer at the 1 fs time step. In addition, the selenium derivatives required a previous preparation step where the parameters and charges were generated by using the antechamber module of AMBER, using the GAFF force field and AM1-BCC method for charges [91].

## 3. Results and Discussion

### 3.1. Synthesis of Target Compounds

We hypothesized that small molecules containing Se-allyl or Se-propargyl functionalities, mimicking active ingredients of natural products, would yield potent antitumor and anti-Alzheimer agents. Thus, a total of 38 new selenocompounds, grouped in two series, were obtained following the synthetic procedure depicted in Figure 1. **Series a** comprises Se-allyl and **series b** consists of Se-propargyl, all of which are decorated with cynnamoyl, hydrocynnamoyl, adamantyl, phenyl, small heterocycles, and NSAID derivatives (see Figure 2).

Compounds were synthesized following three reaction steps represented in Figure 1. This synthesis procedure was previously reported with slight modifications [92]. First, the starting reagent sodium selenide (NaSeH), which was common to all the target compounds, was synthesized in good yields. For that, elemental Se was reduced by NaBH_4_ in water. Once the selenating agent was synthesized, the corresponding acid chloride was added to the reaction mixture to form the corresponding sodium selenoate by a nucleophilic acyl substitution. The yield of this reaction step depends on the acid chloride aqueous solubility. Finally, the target compounds were obtained through a nucleophilic substitution over the allyl/propargyl bromide. Allyl/propargyl bromide are organic reagents with poor aqueous solubility, making the addition of tetrahydrofuran necessary to achieve good yields. The synthetic procedure was carried out in one-pot and yields ranging from 10 to 68% were achieved. The structures of all the synthetic compounds were confirmed using spectroscopic methods (^1^H NMR, ^13^C NMR, and ^77^Se NMR).

### 3.2. In Vitro Biological Evaluation

#### 3.2.1. Antioxidant Activity

One of the common features of cancer and AD is oxidative damage. Therefore, it is feasible to believe that the development of a molecule with antioxidant capacity could be beneficial for the prevention and/or treatment of both diseases. Taking this objective into account, radical scavenging ability of the synthesized compounds were evaluated using the DPPH assay. Firstly, determinations were performed at 0.03 mg/mL in the time range (0, 5, 15, 30, 60 and 90 min). Ascorbic acid and trolox were used as the positive controls. The data are expressed as a percentage of DPPH scavenging activity in at least three independent experiments performed in triplicates (Appendix A). Compound **13a** stood out as the only active compound showing a DPPH scavenging activity of 22.38% at 90 min. Thus, further determinations of compound **13a** were performed at five different concentrations ranging from 3.00 × 10^−4^ to 0.30 mg/mL and were recorded at different time points (15, 30, 60, 90, 120, 150, and 180 min). The data are expressed as the percentage of cell growth ± SD in at least three independent experiments performed in triplicates (Appendix A).

Compound **13a** was able to scavenge the DPPH activity, with values that reached 76% of inhibition at 0.3 mg/mL (see Figure 3). It showed concentration- and time-dependent radical scavenging activity in vitro.

#### 3.2.2. Antiproliferative Activity

As a first approach, all the compounds were tested in vitro against lung (HTB-54), prostatic (DU-145), colon (HT-29), and triple-negative breast (MDA-MB-231) cancer cell lines at two concentrations (50 and 10 µM) for 48 h. Then, the MTT colorimetric assay [67] was used to evaluate cell proliferation after the treatment. The data are expressed as the percentage of cell growth ± SD in at least three independent experiments performed in quadruplicates (Appendix A). Results at 10 µM concentration are summarized in Figure 4.

As shown in Figure 4, some compounds were active under our experimental conditions. These results allowed us to determine some preliminary structure-activity relationships, since:The presence of the propargyl group leads to a lower antiproliferative effect (**series b**) compared to compounds of **series a**, that showed more activity against the four cancer cell lines.Among the compounds of **series a**:
The presence of the adamantyl ring (compound **1a**) instead of a benzene ring (compound **2a**) failed to increase the antiproliferative activity.Among the different substituents of the phenyl ring, the incorporation of the chlorine atom in the “2” position of the ring (compound **3a**) seems to not be important for the inhibition of cell viability, as no significant differences were observed with the unsubstituted phenyl ring compound (compound **2a**). Nevertheless, the presence of the benzodioxol ring (compound **4a**), which is a benzene derivative containing the methylenedioxyl functional group, appears to lose cytotoxic effect in HTB-54, DU-145, and MDA-MD-231 cancer cell lines, whereas it seems to be active in the HT-29 cancer cell line compared to the unsubstituted phenyl ring compound (compound **2a**).The unsaturation of the 2-carbon linker (compound **5a**) between the phenyl ring and the carbonyl group seems to lose the cell inhibitory effect, since the saturated linker leads to a higher antiproliferative activity (compound **6a**). Moreover, no significant differences were observed with the phenyl ring compound directly bonded to the carbonyl group (compound **2a**).The incorporation of a chlorine atom in position “2” of the pyridyl ring seems not to be important for the inhibition of the cell viability, since no significant differences were observed between the chlorine-substituted derivative and the unsubstituted derivative (compounds **8a** and **7a**, respectively).Nevertheless, the incorporation of a chlorine atom in position “3” of the thiophenyl ring (compound **10a**) does appear to increase the inhibition of cell viability compared to the unsubstituted ring (compound **9a**).The presence of a furyl ring instead of a thiophenyl ring leads to increased cytotoxic activity (compounds **9a** and **11a**, respectively). In contrast, the incorporation of the isoxazolyl ring in place of the furyl ring results in less inhibition of cell viability (compounds **11a** and **12a**, respectively).NSAIDs derivatives appear to exert greater anti-proliferative activity than carbo- and hetero-cycle derivatives (compounds **13a**–**19a**). Aspirin, ibuprofen, naproxen, and indomethacin derivatives (compounds **14a**, **15a**, **16a**, and **17a**, respectively) stand out by demonstrating cell viability of less than 55% at 10 µM against the panel of the four cancer cell lines. The mefenamic acid derivative (compound **18a**) showed selectivity against the lung cancer HTB-54 cell line. The flumenamic acid derivative (compound **19a**), however, showed potent anti-proliferative activity against three tumor lines (HTB-54, DU-145, and HT-29) with no activity against the MDA-MD-231 tumor line.

As shown in Figure 4, compounds **14a**–**17a** were found to be the most active, as ones with a reduction of the cell growth greater than 45%, after 48h of treatment at 10 µM in at least 3 of the 4 tumor cell lines. Thus, they were selected to further investigate the cytotoxicity at seven concentrations between 0.1 and 100 µM against HTB-54 (lung), HT-29 (colon), HCT116 (colon), and MCF-7 (breast) cancer cell lines. Moreover, these compounds were also evaluated against mammary gland (184B5) and bronchial epithelium (BEAS-2B) nonmalignant cell lines, and the selectivity indexes (SI) were determined as the ratio of IC_50_ values obtained for nonmalignant cells and the homolog cancer cells. IC_50_ and SI values are shown in Table 3. The selected cancer cell lines displayed sensitivity to the action of these Se-derivatives. In this context, compounds **14a** and **17a** exhibited potent antiproliferative activity with IC_50_ values below 10 µM in all tested cancer cell lines. Interestingly, these Se-NSAID derivatives showed greater antiproliferative activity than their parent drug (Table 3). Thus, the introduction of the selenoester moiety along with the allyl chain in the structure of NSAIDs led to far more potent analogs. However, these compounds exhibited SI values below 1.5 in breast and lung cancer. It is known that high SI values are desirable since they reflect efficacy with less toxicity. Remarkably, these NSAID derivatives that present the allyl chain in their structure, demonstrated slightly better cytotoxic activity than other Se-NSAID derivatives recently published in relation to MCF-7 cells [93].

#### 3.2.3. NCI-60 Analysis of the Compounds **14a** and **17a**

Compounds **14a** and **17a** showed the lowest IC_50_ values against the tested cancer cell lines. Therefore, they were further submitted to the Developmental Therapeutics Program (DTP) of National Cancer Institute (NCI). Both compounds presented promising results in the One Dose (10 µM) assay against a panel of 60 cancer cell lines (Appendix A). Compound **14a** demonstrated a mean growth percent of −5.89 and compound **17a** showed a mean growth percent of −26.70. In most of the cancer cell lines, both molecules presented cytotoxic behavior after 48 h of treatment, highlighting ovarian and CNS cancer cell lines, which are among the most resistant to current treatments. Given the promising results, both compounds were selected to perform five dose assays, in which compounds were tested against the same panel of 60 cancer cell lines for 48 h at 5 different concentrations. Dose-response curves were determined and GI_50_, TGI, and LD_50_ values were calculated for each compound in each cancer cell line. Results are depicted on Figure 5, Appendix A. Both compounds showed average GI_50_ values in the low micromolar range (1.79 µM for **14a** and 1.74 µM for **17a**), lower than other cytotoxic drugs such as gefitinib (3.24 µM), oxaliplatin (2.89 µM), and 5-fluorouracil (57.5 µM) in the same cell line panel [94]. In addition, these compounds were highlighted for the potent antiproliferative activity against the most resistant cancer cell lines of the panel [95]. Compound **14a** showed GI_50_ values of 0.34 µM, 0.49 µM, 1.9 µM, 1.9 µM, and 1.5 µM against NCI-H322M (non-small cell lung), SNB-19 (CNS), SK-MEL-5 (melanoma), OVCAR-3 (ovarian), and OVCAR-8 (ovarian) cancer cell lines, respectively. Likewise, compound **17a** demonstrated GI_50_ values of 0.42 µM, 0.66 µM, 1.7 µM, 1.8 µM, and 1.7 µM against the same cancer cell lines, respectively. These outstanding results emphasize the successful design of these hybrid molecules combining NSAIDs and allylic fragments derived from natural products, to achieve molecules with high antiproliferative activity.

#### 3.2.4. AChE Inhibition

With the aim of developing dual compounds for the treatment of cancer and AD, all compounds were evaluated as inhibitors of AChE enzyme. AChE enzyme has played a major role in AD, since clinical data have demonstrated that the brain of patients with AD have significant neurodegeneration, reduced cholinergic neurons, and a severe deficiency of acetylcholine (ACh) [68]. Therefore, inhibitors of this enzyme have been developed for the treatment of this disease, such as galantamine, donepezil, rivastigmine, and tacrine.

All the synthetic derivatives were tested and commercially available galantamine was used as a reference standard. In the screening at 1 µM (see Table 4), most of the compounds showed a moderate AChE inhibition, with values ranging from 15 to 30%, and no significant differences were observed between allyl and propargyl derivatives. However, compounds **10a**, **13a** and **14a** demonstrated inhibitory activity similar to galantamine, with inhibition values of 42.46%, 51.80%, and 43.46%, respectively. Therefore, their dose-response curves were determined, and they showed IC_50_ values in the low micromolar range (Table 4). Compound **10a**, presenting both the 2-(3-chloro)thiophenyl and allyl fragments, exhibited an IC_50_ value of 2.4 µM, which is comparable to galantamine activity. Compound **13a**, the salicyl and allyl derivative, showed an IC_50_ value greater than galantamine, and compound **14a** (the aspirin and allyl derivative), demonstrated an IC_50_ of 0.9 µM lower than the IC_50_ value of galantamine. Thus, compound **14a** has exhibited similar or even slightly greater AChE inhibitory activity than galantamine. Figure 6 depicts the dose-inhibition curve of compound **14a** and galantamine.

#### 3.2.5. In Vitro Blood–Brain Barrier Permeation Assay

Good penetration through the blood–brain barrier (BBB) is a necessary condition for drugs designed for the treatment of AD. Therefore, the BBB penetration of the compounds that were found to be the most potent in vitro AChE inhibitors (**10a**, **13a**, and **14a**) together with galantamine were tested using the PAMPA-BBB assay [83]. This assay measures the passive diffusion of a compound across a membrane coated with PBL. It is known that compounds with a Pe value greater than 4 × 10^−6^ cm s^−1^ can easily cross the BBB and reach the CNS, whereas compounds with a Pe value below 2 × 10^−6^ cm s^−1^ cannot pass it. Compound **13a** was tested but signs of degradation were shown through the experiment. Compounds **10a** and **14a** showed 3-fold higher Pe values (38.63 × 10^−6^ and 36.92 × 10^−6^ cm s^−1^, respectively) compared to galantamine (12.27 × 10^−6^ cm s^−1^). Moreover, the simulation performed with the preADMET predictor was confirmed by the experimentally obtained data, as both studies suggest higher BBB penetration for compounds **10a** and **14a** compared to galantamine (reference drug). Results are depicted in Table 5.

### 3.3. Molecular Dynamics Simulations

The structure of AChE has been extensively studied since it was first characterized in 1991 [97]. The active site of AChE is not on the surface of the protein but is inside a 20 Å deep gorge lined with many aromatic residues. At the entrance to the gorge is the peripheral anionic site (PAS), where aromatic residues (predominantly tryptophan) interact with cationic ligands. This interaction can also be observed at the catalytic anionic site (CAS), which is located at the base of the gorge. PAS plays a key role in the binding and orientation of acetylcholine into the gorge. Acetylcholine transiently forms a π-cation interaction with tryptophan and the carbonyl of the acetyl group forms a weak hydrogen bridge with tyrosine further down the gorge. These interactions position the acetylcholine towards the active site. At the base of the gorge, a second tryptophan plays a key role in the CAS. Again, acetylcholine forms a π-cation interaction between its quaternary amine and the tryptophan ring. The acetyl group is bound in the acyl pocket, formed from further aromatic residues (Phe295, Phe297, and Trp236) lining the base of the gorge. This binding of acetylcholine to the CAS and the acyl pocket places it in the active site of the enzyme, where three amino acids, glutamate, histidine, and serine, known as the catalytic triad, are located [98,99].

To further investigate the interaction mode of compound **14a** with recombinant human AChE (PDB code: 4BDT), we performed a series of molecular dynamics simulations. The ligand from the crystal was replaced by the allyl derivative prior to the molecular dynamic simulations. As shown in Figure 7, compound **14a** establishes interactions in the PAS of the enzyme, at the entrance to the gorge. Tyr124 in grey color forms two hydrogen bonds with the oxygens from the two carbonyl functional groups (acetyl and selenoester groups) and Trp286 interacts with the aromatic ring through a CH-π interaction. Likewise, Tyr341 interacts with the same aromatic ring from the ligand through a π-π stacking interaction, and the allylic chain lies between Phe338 and Phe297 from the acyl binding pocket.

Given the results obtained in the in vitro inhibition of AChE (Table 4), we decided to perform molecular dynamics studies with compound **14b** as well. Compounds **14a** and **14b** differ only in the presence of a triple (**14b**) or a double (**14a**) terminal bond in the alkyl chain; however, compound **14a** showed higher in vitro AChE inhibition activity, similar to that of galantamine. Therefore, molecular dynamics studies were performed to analyze the interaction of compound **14b** with the enzyme binding site and compare it with compound **14a**. In this study, it was observed that with the majority conformation of compound **14b,** the interactions with the protein were much weaker than in the case of compound **14a**. As can be shown in Figure 7C, hydrogen bonds between the carbonyl groups and Tyr124 of PAS was not found, and only very weak CH-π or π-π stacking interactions were observed with Tyr124, Tyr341, and Trp286 (green) of PAS and the aromatic ring of the ligand. It was also revealed that the acyl binding site remains free (purple) for interaction with acetylcholine, which would explain why compound **14b** is not a good inhibitor of AChE.

All these data seem to suggest that compound **14a** prevents the interaction of acetylcholine with the aromatic residues of the PAS, which is critical for the molecule to settle into the active site. In addition, it appears that the allylic chain plays a key role, as it would hinder the interaction of the acyl binding site residues with acetylcholine, preventing its proper placement in the CAS and the action of the catalytic triad.

## 4. Conclusions

Overall, **14a** was identified as a dual anticancer and AChE inhibitory agent, showing a concentration-dependent inhibition. For example, to exert its AChE inhibitory capacity, it exhibited an IC_50_ value of 0.9 μM and for this concentration it showed no antiproliferative activity. In addition, it has been observed that the introduction of the allyl fragment, which is present in natural products such as garlic, as opposed to the propargyl fragment, provides antiproliferative activity, AChE inhibitory activity, and a slight radical scavenging ability. Compound **14a** is the first Se-containing agent with dual in vitro antiproliferative and AChE inhibitory activities reported so far. The introduction of Se into molecules with potential AChE inhibitory activity is a novel strategy currently under development. Thereby, scientific references are scarce. Ebselen has been demonstrated to inhibit AChE activity with an IC_50_ value of 29 µM, which is 32-fold higher than the IC_50_ presented by compound **14a** [100]. However, the compound selenepezil, a derivative of the fusion of donepezil and ebselen, exhibited potent AChE inhibition with an IC_50_ value of 0.097 µM (9-fold lower than the IC_50_ value of **14a**). Additionally, selenepezil mimicks the activity of GPx and exhibits scavenging activity towards radical species generated by hydrogen peroxide in vitro [100].

On the other hand, no relation was found between radical scavenging ability and AChE inhibitory activity or antiproliferative activity. For anticancer activity, this was to be expected, since it has been observed that molecules with antioxidant capacity do not provide any benefit during the treatment of the disease, but rather worsen it by protecting the tumor cells from oxidative damage and, hence, promoting their progression. However, molecules with antioxidant activity might be favorable in the prevention of the disease. Previously, our research group has reported the synthesis of Se-molecules with antioxidant and cytotoxic activity in vitro, highlighting the acylselenourea and selenourea derivatives [36,73]. Therefore, a new approach could be to substitute the selenoester group of the synthesized molecules by the acylselenourea or selenourea group and study their antioxidant and antiproliferative capacity. On the other hand, ebselen, which acts as a glutathione peroxidase mimic, is widely known because of its potent antioxidant capacity [43].

Altogether, the work presented herein warrants future studies to further assess the in vivo efficacy, toxicity, and characterization of the mechanism of action of compound **14a** in both cancer and AD models.

## Data Availability

The data presented in this study are available on reasonable request from the corresponding author.

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
