# Peer review of "Seleno-Analogs of Scaffolds Resembling Natural Products a Novel Warhead toward Dual Compounds"

_antioxidants, 2023, doi:10.3390/antiox12010139_

Round 1

Reviewer 1 Report

Comments and remarks can be found in the attached pdf file.

Author Response

The authors do not write on the basis of which literary sources they developed a scheme for the synthesis of
target products. If this is their own development, then it is, to put it mildly, far from perfect. If the authors
tried to work out the synthesis methodology on the basis of some analogies, then its implementation in the
version presented in the manuscript does not seem very professional. Indeed, from the table of yields of
compounds 1a,b – 19a,b given here, about a third of the synthesized compounds (11 out of 38) have a yield
of 10%, and more than 70% do not reach the yield of 30%! Moreover, the cases seem incomprehensible
when, in almost identical experiments (minimum difference in the structures of acid chlorides), the yields of
allyl and propargyl products (examples for compounds 7a,b and 8a,b) are completely opposite? In this case,
how does such an inverse combination of yields agree with the statement of the authors (line 646) about the
effect of the solubility of the acid chloride in aqueous solutions?

Response: We truly and deeply acknowledge the reviewer for the thorough revision and the fruitful
comments which have helped to increase the quality of the manuscript. The synthesis procedure has been
based on a previously reported one (Molecules 2009, 14, (9), 3313-38), so the corresponding citation has
been added to the manuscript (line 370). This synthetic procedure was chosen because it is the method of
reference used in our research group for the synthesis of selenoesters (Eur J Med Chem 2022, 244, 114839;
Eur J Med Chem 2014, 73, 153-66).
In the case of yields, it also caught our attention. As you rightly comment, the synthesis procedure of both
series is almost similar, so we should not observe such a marked difference in the yields. However, during the
purification of the compounds, we encountered several side products that could perhaps explain these yields.
Based on our experience, the number of side products also increase when using ethanol as reaction solvent.
Thus, we decided to use the aqueous reaction media. Additionally, we have found very interesting and
unexpected seleno-containing products that are currently under investigation and will be communicate
shortly.
In this regard, the following questions arise:
1) Why do the authors not use an organic rather than an aqueous medium, for example, EtOH, to obtain
NaHSe?
Response: We acknowledge the reviewer for this observation. In view of the problems with the yields, the
synthesis of the compounds was carried out using EtOH as reaction medium, 0 °C, and using a nitrogen
atmosphere in the first step to avoid the possible oxidation of the intermediate. We performed this method of
synthesis on several occasions and the problem we encountered was that the major product of the reaction
was the ester formed between the acid chloride and ethanol (which being the reaction medium was in great
excess). Therefore, we had to discard these conditions.
2) There are examples in the literature when the preparation of the target allyl derivatives of selenoates
described by the authors was carried out in several stages, by the sum of the stages giving a total yield of the
final derivative of over 80%! Thus, at the first stage, according to the scheme, potassium arylselenobenzoate
was obtained, which at the next stage was introduced into the reaction with allyl bromide, forming almost
quantitatively a homologue of the product, which was obtained by the authors of this manuscript with a
yield of 10%.

Response: We deeply acknowledge the reviewer for this insightful comment. It is a synthetic procedure that
we will keep in mind for future projects, and we will try to see if we can improve the yields. As mentioned
above, the use of EtOH as a reaction medium led us to obtain the ester. However, we will keep it in mind for
future projects.
Lines 157-158: There is no confirmation of the chemical composition of the isolated target compounds either
by elemental analysis or by HRMS spectroscopy data. The correspondence of the structure of the obtained
compound to the structure of the proposed product was made only on the basis of NMR spectroscopy. In
recent articles by these authors (refs 36, 80, 87), they cited HRMS data as proof of the brutto-formula of new
organoselenium compounds.
Response: We thank the reviewer for this comment. We believe that on this case the characterization and
purity of the compounds by NMR is enough information to be able to assess both structure and purity of the
compounds. Nowadays, many journals no longer ask for chemical composition analysis. Instead, they do ask
for purity analysis by 1 H q-NMR or HPLC.
Lines 172-173: It is not specified which eluent was used for column chromatography, and also the Rf range
for isolated products.
Response: We acknowledge the reviewer for this useful observation. The following information has been
added in lines 194- 196: A gradient of hexane/ethyl acetate, ranging between 0 % and 50 % of ethyl acetate
was used as eluent. The mobile phase of the TLCs was hexane/ ethyl acetate with a ratio of 9:1. The Rf range
was from 0.18 to 0.75.
Line 179 (187, 194, 202, etc): In compound names, a space is placed after Se-allyl; Line 348 (355, 361, 368
etc.): It is better to replace Se-(prop-2-in-1-yl) with Seprop-2-ynyl and then insert a space to separate the
prefix from the main compound name 1b-19b
Response: We thank the reviewer for this useful suggestion. The names of the compounds of series a and
series b have been modified according to the comments made. The chemical characterization of the
synthesized compounds has been reorganized in two tables (Table 2 for series a, and Table 3 for series b) as
suggested by reviewer 3.
Line 185: 2 Signals in 13 С NMR are very close, 28.3 and 26.6 ppm. On the basis of what reasoning did the
authors attribute one to СAd, and the other to CH 2 Se?
Response: We thank the reviewer for this useful comment. This assignment was made based on the 13 C NMR
of its b-series analogue, where C Ad appears at 28.5 ppm.
Lines 227-228 and 274-275: “was obtained” is written twice; Line 251: 77Se NMR…  (ppm) : 232 should be
changed to 557; Line 294: orangey; Line 567: …bis(2-nitrobenxoic acid);
Response: We acknowledge the reviewer for these observations. All of them has been corrected.

Line 498: If the authors refer to a well-known method (described by Svinyarov), a literary reference to the
technique should be given
Response: We thank the reviewer for this comment. The corresponding reference has been added in line 208.
Line 580 and 583: PBS (what is a full name?)

Response: We thank the reviewer for this observation. PBS refers to phosphate buffered saline. The full name
has been added in line 307. Moreover, at the request of reviewer 3 a list of abbreviations has been included.

Line 653: It should be better to draw R 3 CH 2 CH=CH 2 and CH 2 CCH ; Line 655: H 2 O (not H2O); Line 656:
BrCH 2 CHCH 2 or BrCH 2 CC have to changed to BrCH 2 CH=CH 2 or BrCH 2 CCH; Line 670: At 5 min records were
made too.
Response: We acknowledge the reviewer for these corrections. All of them have been made.

Reviewer 2 Report

In this study, a total of 39 Se-derivatives were synthesized, purified, and spectroscopically characterized by RMN. Thier antioxidant and  antiproliferative abilities were tested in breast, lung, prostate, and colorectal cancer cells.  The experiment was well desgisned. The following  changes could improve the quality of the paper.

1. remove the '.' from the title.

2. please do not use blod style for '13a', '14a', '17a' etc. in the abstract.

3. introduce the antioxidant and  antiproliferative abilities of previouse stuidied Seleno-anlogs. Waht is novel for your 39 Seleno-anlogs?

4. The presentation of the Figures 2 and 3 are clear.

5. Please add the replicates (n=?) in the footnote and Figure lengends.

6. The aurthors need to compare the efffcts of the synthesized Seleno-anlogs with the previouse selenium compounds.

Author Response

In this study, a total of 39 Se-derivatives were synthesized, purified, and spectroscopically characterized by
RMN. Thier antioxidant and  antiproliferative abilities were tested in breast, lung, prostate, and colorectal
cancer cells.  The experiment was well desgisned. The following  changes could improve the quality of the
paper
1. remove the '.' from the title.
2. please do not use blod style for '13a', '14a', '17a' etc. in the abstract
Response: We acknowledge the reviewer for these observations. Both have been made.
3. introduce the antioxidant and  antiproliferative abilities of previouse stuidied Seleno-anlogs. Waht is novel
for your 39 Seleno-anlogs?
Response: We thank the reviewer for this useful suggestion. New information regarding previous Se analogs
has been added in lines 143-153.
4. The presentation of the Figures 2 and 3 are clear.
Response: We deeply acknowledge the reviewer for this observation
5. Please add the replicates (n=?) in the footnote and Figure lengends.
Response: We thank the reviewer for this useful suggestion. Replicates information has been included in the
footnotes and legends of Fig. 2 and 3 and Table 6.
6. The aurthors need to compare the efffcts of the synthesized Seleno-anlogs with the previouse selenium
compounds.
Response: We deeply acknowledge the reviewer for this insightful comment. Information comparing the
synthesized Se-analogs with previous Se compounds has been added in lines 722-731 and 738 – 743.

Reviewer 3 Report

Dear authors. The work is really impressive. I give only several proposals to improve the perception of the data and indicate several misprints.

Major comments

1)     Introduction should be expanded as not only garlic Se-compounds but also selenoproteins and selenoaminoacids are known to produce a positive effect in Alzheimer and cancer treatment. May be the below citations will help in the decision of this question:

-Aaseth J, Alexander J, Bjørklund G, Hestad K, Dusek P, Roos PM, Alehagen U. Treatment strategies in Alzheimer's disease: a review with focus on selenium supplementation. Biometals. 2016 Oct;29(5):827-39. doi: 10.1007/s10534-016-9959-8.

- Zhang Z-H and Song G-L (2021) Roles of Selenoproteins in Brain Function and the Potential Mechanism of Selenium in Alzheimer’s Disease. Front. Neurosci. 15:646518. doi: 10.3389/fnins.2021.646518

2)     it is desirable to indicate all abbrevitions in a separate list

3)     Material and Methods section is difficult to read. Is it possible to compose a Table of results, including: yield, NMR spectrum data and purity (1H NMR, 13C NMR, and 77Se NMR), etc? With two groups of compounds:‘a’ and ‘b’

4)     line 625- what was the necessity to use Kelvin ’K” for temperature, why not Celsius ‘C’?

5)     line 643- indicate what was the size of Se particles

6)     It is highly desirable to compare AOA of the synthetized compounds with the known Se- derivatives, like ebselen

Minor comments:

1.      Line 150 TLC- what system of solvents was used?

2.      Line 509 ‘was prepared daily and protect from light’= was protected

3.      Line 597 ‘Every sample were analyzed’- was

4.      what is ‘fs’ abbreviation

5.      Line 642 ‘NaHSe’ better write ‘NaSeH’

6.      Lines 700-730- of high value the relationship between the structure and biological activity

7.      Line 760 ‘theses' change to ‘these’

8.      Line765 ‘leds’- change to  ‘led’

Author Response

Major comments
1)     Introduction should be expanded as not only garlic Se-compounds but also selenoproteins and
selenoaminoacids are known to produce a positive effect in Alzheimer and cancer treatment. May be the
below citations will help in the decision of this question:
-Aaseth J, Alexander J, Bjørklund G, Hestad K, Dusek P, Roos PM, Alehagen U. Treatment strategies in
Alzheimer's disease: a review with focus on selenium supplementation. Biometals. 2016 Oct;29(5):827-39.
doi: 10.1007/s10534-016-9959-8.
- Zhang Z-H and Song G-L (2021) Roles of Selenoproteins in Brain Function and the Potential Mechanism of
Selenium in Alzheimer’s Disease. Front. Neurosci. 15:646518. doi: 10.3389/fnins.2021.646518
Response: We thank the reviewer for this suggestion. We believe this information is relevant to the
manuscript; therefore, it has been included in lines 95-104 and 108-111.
2)     it is desirable to indicate all abbrevitions in a separate list
Response: We acknowledge the reviewer for this observation. An abbreviations list has been added before
the references.
3)     Material and Methods section is difficult to read. Is it possible to compose a Table of results, including:
yield, NMR spectrum data and purity (1H NMR, 13C NMR, and 77Se NMR), etc? With two groups of
compounds:‘a’ and ‘b’
Response: We thank the reviewer for this useful comment. Therefore, the information related to the
characterization of the compounds has been reorganized in two tables, table 2 for the series a and table 3 for
the series b.
4)     line 625- what was the necessity to use Kelvin ’K” for temperature, why not Celsius ‘C’?
Response: We thank the reviewer for this observation. The Molecular Dynamics simulations were run by
using the AMBER suite software. This program requires to input the temperature in Kelvin units. This is the
reason why Kelvin units are used.
5)     line 643- indicate what was the size of Se particles
Response: We acknowledge the reviewer for this comment. The elemental selenium used in the synthetic
procedure is a powdered grey selenium purchased from Sigma-Aldrich (CAS 7782-49-2) in which the particle
size is not indicated.
6)     It is highly desirable to compare AOA of the synthetized compounds with the known Se- derivatives, like
ebselen
Response: We deeply acknowledge the reviewer for this insightful suggestion. Information comparing the
synthesized Se-analogs with previous Se compounds has been added in lines 722-731 and 738 – 743.

Minor comments:
1. Line 150 TLC- what system of solvents was used?
Response: We acknowledge the reviewer for this useful suggestion. The following information has been
added in lines 194- 196: A gradient of hexane/ethyl acetate, ranging between 0 % and 50 % of ethyl acetate
was used as eluent. The mobile phase of the TLCs was hexane/ethyl acetate with a ratio of 9:1. The Rf range
was from 0.18 to 0.75.
2.      Line 509 ‘was prepared daily and protect from light’= was protected
3.      Line 597 ‘Every sample were analyzed’- was
5.      Line 642 ‘NaHSe’ better write ‘NaSeH’
7.      Line 760 ‘theses' change to ‘these’
8.      Line765 ‘leds’- change to  ‘led’
Response: We eagerly acknowledge this set of comments. All of them have been made.
4.      what is ‘fs’ abbreviation
Response: We thank the reviewer for this comment. ‘fs’ means femtosecond which has been included in the
abbreviations list. It refers to the unit of time used for the time step.
6.      Lines 700-730- of high value the relationship between the structure and biological activity
Response: We thank the reviewer for this observation.

Round 2

Reviewer 1 Report

I would like to thank the authors for their careful attention to the comments and corrections they made. Considering the importance of the research topic presented and the interesting results obtained by the authors related to the biological activity of new organoselenium compounds, I am not interested in delaying the process of accepting a manuscript for publication. For my part, as a chemist specializing in the methodology of organic synthesis, I would like to recommend the authors to continue to try to find acceptable methods for the synthesis of target organoselenium compounds that would give satisfactory yields of desired products.